# Transcriptomic Analysis of the Effect of Pruning on Growth, Quality, and Yield of Wuyi Rock Tea

**DOI:** 10.3390/plants12203625

**Published:** 2023-10-20

**Authors:** Qi Zhang, Ying Zhang, Yuhua Wang, Jishuang Zou, Shaoxiong Lin, Meihui Chen, Pengyao Miao, Xiaoli Jia, Pengyuan Cheng, Xiaomin Pang, Jianghua Ye, Haibin Wang

**Affiliations:** 1College of Tea and Food, Wuyi University, Wuyishan 354300, China; zhangqi1113@126.com (Q.Z.);; 2College of Life Science, Fujian Agriculture and Forestry University, Fuzhou 350002, Chinaa13883402655@163.com (J.Z.); 3College of Life Science, Longyan University, Longyan 364012, China

**Keywords:** gene expression, growth indicator, meizhan (*Camellia sinensis*), pruning, quality, transcriptomics

## Abstract

Pruning is an important agronomic measure in tea plantation management. In this study, we analyzed the effect of pruning on gene expression in tea leaves from a transcriptomics perspective and verified the results of a transcriptomic analysis in terms of changes in physiological indicators of tea leaves. The results showed that pruning enhanced the gene expression of nine metabolic pathways in tea leaves, including fatty acid synthesis and carbohydrate metabolism, nitrogen metabolism, protein processing in the endoplasmic reticulum, and plant hormone signal transduction, thereby promoting the growth of tea plants and increasing tea yield. However, pruning reduced the gene expression of nine metabolic pathways, including secondary metabolites biosynthesis, flavonoid biosynthesis, phenylpropanoid biosynthesis, and sesquiterpenoid and triterpenoid biosynthesis, and lowered the content of caffeine, flavonoids, and free amino acids in tea plant leaves. In conclusion, pruning could promote the growth of tea plants and increase the yield of tea, but it was not conducive to the accumulation of some quality indicators in tea leaves, especially caffeine, flavonoids, and free amino acids, which, in turn, reduced the quality of tea. This study provides an important theoretical reference for the management of agronomic measures in tea plantations.

## 1. Introduction

Tea plants are an important economic crop, and the number of the world’s tea-producing countries and regions reached more than 60 in 2022, with a tea-drinking population of more than 2 billion. China is the largest tea producer and consumer, accounting for 47% of global tea production. In 2021, total tea production in China reached 3.08 million tons, with the total output value of dried gross tea exceeding CNY 290.9 billion, driving 80 million tea farmers to increase their incomes and become rich [1]. It can be seen that tea plays an important role in promoting China’s agricultural and rural economic development and rural revitalization.

Fujian Province is the main tea-producing region in China, and Wuyi rock tea is produced in Wuyi Mountain, Fujian Province, which is unique because of the particularity of its planting environment. The production process of Wuyi Rock Tea is semi-fermented according to the degree of fermentation, Wuyi Rock Tea belongs to a kind of oolong tea. At present, the main varieties of tea plants in Wuyi Mountain are basically shrubs. It is well known that tea plants require a lot of manual management in the planting process to improve tea yield and quality, including irrigation, fertilization, weeding, pest control, pruning, and other agronomic measures [2,3,4,5]. Irrigation, fertilization, weeding, and pest control were the agronomic measures necessary for the cultivation and management of tea plants, while pruning was somewhat different for different varieties of tea plants, especially shrub tea plants [6]. Pruning was an essential agronomic measure in the cultivation of shrub tea plants. This was because pruning was beneficial in stimulating the growth of lateral buds of tea plants, thus increasing tea yield; at the same time, pruning could reduce labor in tea harvesting and improve tea harvesting efficiency [7,8]. Accordingly, many scholars have conducted a large number of studies on the effect of pruning on the growth of tea plants. For example, Bora et al. [9] found that long-term pruning would alter the microbial community structure of the rhizosphere soil of the tea plant, and the tea plant would selectively promote the growth and colonization of important functional microorganisms for its own growth. Sarmah et al. [10] found that pruning was beneficial in promoting tea plants to absorb more trace elements from the soil. Jiang et al. [11] analyzed the rhizosphere soil of the tea plant and found that pruning could alleviate the degradation of the rhizosphere soil of the tea plant, improve the ecosystem function of the tea plantation, and thus stabilize tea yield, but it would reduce the content of polyphenols and amino acids in the tea leaves. Borgohain et al. [12] analyzed soil litter accumulation and found that tea plant litter after pruning could promote tea plant growth and improve tea plant biomass. It can be seen that pruning was indeed beneficial in promoting the growth of tea plants.

Pruning is also a form of stress for the tea plant itself. It has been reported that plants under stress stimulate secondary messengers to activate cellular signaling, which, in turn, triggers transcriptomic responses related to plant defense, allowing plants to adopt different strategies to cope with the effects produced by different stresses [13,14]. Jia et al. [15] found that the gene expression of tea plant leaves changed under abiotic stress, and this change affects the synthesis of compounds in tea plant leaves. Zhang et al. [16] also found that tea plants in abiotic stress will resist environmental stress by increasing their own growth capacity. It can be seen that after pruning, the tea plant may change its own functions in order to adapt to the environment. Our research team’s previous study found that the enzyme activity and microbial functional diversity of the rhizosphere soil of the tea tree changed significantly after pruning, and this change improved the nutrient conversion capacity of the soil, which promoted the nutrient uptake capacity of the tea tree and then promoted the growth of the tea tree [17]. At the same time, it was found that pruning, although it promotes the growth of tea trees, is not conducive to the synthesis of compounds related to tea quality [18]. The synthesis of compounds related to tea quality is closely related to the intensity of the corresponding metabolite pathways and gene expression in tea tree leaves. How the tea tree responds to the effects of pruning and how it regulates the expression of genes within the tea tree to affect its growth and tea quality has rarely been reported. Accordingly, in this study, we analyzed the effect of pruning on the gene expression of tea plant leaves from the perspective of transcriptomics using Meizhan (*Camellia sinensis* cv. Meizhan), the main tea plant variety grown in Wuyi Mountain, and verified the results of the transcriptomics analysis in terms of the changes in leaf physiological indicators, with a view of providing a certain reference for the cultivation and management of tea plants. 

## 2. Results and Discussion

### 2.1. Transcriptomic Analysis of Tea Leaves

After Illumina sequencing, a total of 251,514,572 raw reads were obtained from the six libraries. In total, 243,980,532 clean reads were obtained after filtration, and the percentage of clean reads relative to raw reads in each library was over 99.97% (Appendix A), indicating that transcriptome sequencing results were of reliable quality. Comparison analysis showed that the percentage of clean reads that matched the reference genome was above 87% (Appendix A). It can be seen that the reference genome was well assembled, the measured species were consistent with the reference genome, and there was no contamination in the relevant experiments, so the results could be used for further analysis. Secondly, this study further analyzed the similarity of the three biological replicates in transcriptome sequencing results, and the results showed (Figure 1) that the R^2^ values of the three biological replicates of the unpruned tea plant (MC) transcriptome were above 0.991, and the R^2^ values of the pruned tea plant (MP) transcriptome were above 0.988. It can be seen that the transcriptome data of different samples in this study were repeatable and reliable for in-depth analysis.

### 2.2. Gene Expression Analysis and Key Gene Screening in Tea Leaves under Different Treatments

The results of overall gene expression levels of pruned (MP) and unpruned (MC) tea leaves showed (Figure 2A) that there was no significant difference between MP and MC in overall gene expression levels (*p* = 0.92). The results of PCoA analysis showed (Figure 2B) that the two components could effectively distinguish different samples, with the contribution rate of PCoA1 at 73.93%, PCoA2 at 13.64%, and the overall contribution rate at 87.57%. It can be seen that there may be significant differences between MP and MC in gene expression. Accordingly, the volcano plots were further used in this study to analyze changes in gene expression between MP and MC, and the results showed (Figure 2C) that compared with MC, 514 genes in MP were upregulated, 788 genes were downregulated, and 41,776 genes were not significantly different in expression. It can be seen that the gene expression of tea leaves changed significantly after pruning.

OPLS-DA can be used to establish the correlation model between gene expression and samples and to screen for key genes that can characterize sample differences by variable importance projection value (VIP value) [19]. Meanwhile, to test the reliability of the OPLS-DA model, permutation testing was usually used to verify the model, which evaluated the model’s accuracy [20]. Therefore, in order to screen and obtain key genes that had significant changes in gene expression of tea leaves after pruning, this study further adopted the OPLS-DA model for analysis, and the results showed (Figure 2D) that the R2Y value for the goodness of fit of the OPLS-DA model was 1 (*p* < 0.005), and the Q2 value for predictability was 0.994 (*p* < 0.005). It can be seen that both R2Y and Q2 values of the model reached significant levels, indicating that the model had a good degree of fit and high reliability and could be used for further analysis. The results of the OPLS-DA scoring map showed (Figure 2E) that OPLS-DA could effectively distinguish MP and MC in different regions. There were significant differences in gene expression between MP and MC. The results from S-plot analysis showed (Figure 2F) that 712 key genes (VIP > 1) differentiated MP from MC, of which 286 key genes were upregulated, and 426 key genes were downregulated in MP compared to MC. It can be seen that the gene expression of tea leaves changed significantly after pruning.

### 2.3. KEGG Pathway Enrichment and Functional Analysis of Key Genes

On the basis of the above analysis, this study further enriched KEGG metabolic pathways of key genes, and the results showed (Figure 3A) that the enrichment of KEGG metabolic pathways for key genes involved 55 metabolic pathways, 19 of which reached significant levels (*p* < 0.05). The gene expression of 19 metabolic pathways was further analyzed, and the results showed (Figure 3B) that after pruning, the gene expressions of 9 metabolic pathways in tea leaves were significantly upregulated (carotenoid biosynthesis, biosynthesis of unsaturated fatty acids, fatty acid biosynthesis, protein processing in endoplasmic reticulum, nitrogen metabolism, pentose, and glucuronate interconversions, starch and sucrose metabolism, amino sugar and nucleotide sugar metabolism, and plant hormone signal transduction), while the other 9 metabolic pathways were significantly downregulated (biosynthesis of secondary metabolites, peroxisome, metabolic pathways, flavonoid biosynthesis, phenylpropanoid biosynthesis, sesquiterpenoid and triterpenoid biosynthesis, alpha-linolenic acid metabolism, tyrosine metabolism, and glycolysis/gluconeogenesis), and the gene expressions of 1 metabolic pathway were not significantly different (Stilbenoid, diarylheptanoid and gingerol biosynthesis).

Pruning is abiotic stress for the tea plant itself, and under abiotic stress, the tea plant itself responds accordingly. Phytohormones were key signaling compounds that regulated growth, development, and response to environmental stress, and when plants were affected by the environment, phytohormone synthesis was rapidly initiated, and signals were transmitted to activate plant defense mechanisms [21]. This study found that the gene expression capacity of plant hormone signal transduction was enhanced after the pruning and carotenoid biosynthesis, biosynthesis of unsaturated fatty acids, fatty acid biosynthesis, and pentose and glucuronate interconversions were also enhanced. Carotenoid biosynthesis pathways have been reported to be enhanced in plants under environmental stress, which, in turn, increases carotenoid content [22]. Fatty acids were beneficial to plant resistance to environmental stress, and the higher their content, the stronger plant resistance to environmental stress [23]. The increased capacity of pentose and glucuronate interconversions was beneficial in improving carbohydrate and energy metabolism, which, in turn, promoted plant metabolism and enhanced plant resistance to environmental stress [24]. It can be seen that after pruning, the tea plant activated the hormone signal transduction mechanism to improve its resistance to environmental stress in order to adapt to the environment. Secondly, plants need energy to synthesize hormones, fatty acids, and other compounds in response to environmental stress. This study found that after pruning the tea plant, the gene expression of protein processing in the endoplasmic reticulum, nitrogen metabolism, starch and sucrose metabolism, and amino sugar and nucleotide sugar metabolism were enhanced. It has been reported that environmental stress favored an increased capacity for starch and sucrose metabolism, amino sugar and nucleotide sugar metabolism, enhanced photosynthesis, and the accumulation of carbohydrates to meet the energy supply for compounds synthesis of plants, especially under abiotic stress [25,26]. In addition, the increased capacity of nitrogen metabolism contributes to nutrient conversion and hormone synthesis in plants [27]. The enhanced protein processing capacity of the endoplasmic reticulum accelerated the transport of lipid and carbohydrate compounds [28]. It can be seen that after pruning, tea plants could improve their accumulation and metabolism of sugar substances, accelerate the synthesis and transport of plant hormones and fatty acids, and improve their adaptability to the environment.

Moreover, this study found that after pruning, metabolic pathways, biosynthesis of secondary metabolites, flavonoid biosynthesis, phenylpropanoid biosynthesis, sesquiterpenoid, and triterpenoid biosynthesis, and alpha-linolenic acid metabolism were all significantly downregulated in gene expression. It has been reported that the main product of alpha-linolenic acid metabolism was linolenic acid, which was catabolized to jasmonic acid, and jasmonic acid was conducive to inducing and enhancing the synthesis of products of secondary metabolic pathways such as phenylpropane, terpenoid and flavonoid [29,30,31]. Plants responded to external environmental stresses in two ways, biotic and abiotic, and there were differences in the ways plants responded to different stresses. Under biotic stress, plants usually resist the external environment by increasing the synthesis of secondary metabolites, especially flavonoids, alkaloids, and terpenoids [32]. Under abiotic stress, plants typically resist the external environment by increasing photosynthesis to promote growth and improved resistance [33,34]. It is evident that pruning was an abiotic stress for tea plants, and after pruning, tea plants did not rely on enhancing the synthesis of secondary metabolites to resist the external environment. Secondly, this study found that the gene expression of tyrosine metabolism, peroxisome, and glycolysis/gluconeogenesis in tea plant leaves was significantly upregulated after pruning. It has been reported that tyrosine plays an important role in hormone synthesis and signal transduction in plants, and tyrosine could induce the accumulation of secondary metabolites, such as phenols and flavonoids [35,36]. Peroxisome could oxidize fatty acids and reduce their content in plants [37]. Glycolysis/gluconeogenesis was a way of sugar catabolism or synthesis without oxygen and was generally initiated and enhanced by plants under heavy stress [38]. It can be seen that after pruning, the secondary metabolic capacity of tea plants decreased, especially in the phenylpropane metabolic pathway, terpenoid metabolic pathway, and flavonoid metabolic pathway, which might lead to a reduction in their ability to synthesize these compounds.

### 2.4. Analysis of the Growth and Quality Indicators of Tea Plants and Their Interactions with Gene Expression of Metabolic Pathway

Based on transcriptome analysis, the growth indicators of the tea plant were further determined in this study, and the results showed (Figure 4A, Appendix A) that leaf area, hundred-bud weight, chlorophyll content, and tea plant yield were significantly increased after pruning (*p* < 0.05). Visible pruning was beneficial to promote the growth of tea plants and thus increased the yield of tea. Interaction analysis showed (Figure 5A) that tea plant growth indicators were positively correlated with the gene expression of pentose and glucuronate interconversions, biosynthesis of unsaturated fatty acids, nitrogen metabolism and fatty acids biosynthesis, starch and sucrose metabolism, carotenoid biosynthesis, plant hormone signal transduction, protein processing in the endoplasmic reticulum, amino sugar, and nucleotide sugar metabolism. The results also confirmed the above transcriptomic findings that tea plants responded to abiotic stress by improving their growth ability and resistance to the external environment after pruning. The results of the analysis of tea quality indicators showed (Figure 4B, Appendix A) that after pruning, the content of tea polyphenols and theanine did not change significantly, but the content of caffeine, flavonoids, and free amino acids decreased significantly, while the content of soluble sugar increased significantly. The results of the correlation analysis showed that (Figure 5B) the content of flavone, caffeine, and free amino acid was negatively correlated with the content of theanine, soluble sugar, and tea polyphenol; secondly, the gene expressions of biosynthesis of secondary metabolites, peroxisome, metabolic pathways, flavonoid biosynthesis, phenylpropanoid biosynthesis, sesquiterpenoid and triterpenoid biosynthesis, alpha-linolenic acid metabolism, tyrosine metabolism, glycolysis/gluconeogenesis were positively correlated with the content of flavone, caffeine, and free amino acid, and was negatively correlated with the content of tea polyphenol, soluble sugar, and theanine. Caffeine and flavonoids were secondary metabolites of plants, mainly from the phenylpropane metabolic pathway and flavonoid metabolic pathway [39]. There were 16 kinds of free amino acids in tea leaves, of which the synthesis framework of 12 amino acids was mainly derived from glycolysis metabolites [40]. In this study, we found that after pruning, the gene expression of biosynthesis of secondary metabolites, phenylpropanoid biosynthesis, flavonoid biosynthesis, and glycolysis/gluconeogenesis were all downregulated. The results of quality indicators also verified that pruning was beneficial to improving the accumulation of carbohydrate substances in tea leaves and promote the growth of tea plants but reduced the synthesis capacity of secondary metabolites, especially the content of caffeine, flavonoids, and free amino acids in tea leaves, which, in turn, reduced the quality of tea. After pruning the tea tree, attention should be paid to the changes in secondary metabolic pathways of the tea tree, especially the changes in phenylpropane and flavonoid metabolic pathways, to avoid the reduction of the synthesis capacity of secondary metabolites of the tea tree leaves due to pruning and to reduce the quality of the tea leaves.

## 3. Materials and Methods

### 3.1. Tea Plantation and Sample Collection 

The experimental site was Foguoyan tea plantation located in the Wuyi Mountain Scenic Area (117.99° E, 27.72° N) at an altitude of 239 m. The tea plant variety was Meizhan (*Camellia sinensis* cv. Meizhan), with a planting area of about 0.8 ha. The tea plants were 8 years old and had not been pruned before. In August 2021, the test area was evenly divided into 6 areas, three of which had tea plants pruned for treatment (MP), i.e., three replicates, and the other three areas had tea plants not pruned, i.e., three control replicates (MC). Tea plants were pruned by trimming 3–5 cm from the tea plant canopy, and pruning litter was left on the surface of the original tea plant soil. 

In October 2021, 700 kg/ha of compound fertilizer (N:P:K = 21:8:16) were applied, respectively. Other tea plantation management measures were treated the same for both pruned and unpruned treatments. The basic conditions of the tea plantation during the experiment are shown in Appendix A. In April 2022, tea plant growth indicators were determined, and tea leaves were collected for leaf transcriptome analysis and tea quality indicator determination [10,17]. Tea leaves were sampled by randomly selecting 10 tea plants treated with MP or MC, respectively, and collecting the inverted second leaves (first mature leaves) of tea plants uniformly and mixing them for a replication sample. The tea leaves collected were immediately stored in liquid nitrogen for transcriptome analysis and quality indicator determination.

### 3.2. Determination of Tea Plant Growth Indicator

Leaf area, hundred-bud weight, chlorophyll content of tea plants, and tea yield in pruned and unpruned areas were measured [41]. Leaf area was determined by randomly selecting 20 mature fresh leaves of the tea plant in each area of the tea plantation using the S-sampling method and measuring the leaf length (L) and width (W), calculating the leaf area of each leaf according to L × W × 0.7, and taking the average value, i.e., one replicate. Hundred bud weight was determined by randomly selecting 100 standard bud tips with resident bud and 3 leaves in each area, weighing them, repeating the randomly selected and measured three times, and taking the average value as a replicate. Tea yield was determined by randomly selecting three 40m^2^ in each area of the tea plantation and picking them according to the traditional tea picking standard (three leaves and one core), taking the average, i.e., one replicate, and converting it to the yield of fresh tea leaves per hectare of the tea plantation. Chlorophyll content was determined by randomly selecting 10 mature leaves from the tea plantations in each area, using a chlorophyll detector (LD-YD, Shenzhen, China), and taking the average value, which was one replicate. During the experiment, there were three independent areas for both pruned and unpruned treatments and three independent replicates for the determination of each indicator.

### 3.3. Determination of Tea Quality Indicator

The quality indicators of tea leaves were mainly determined by the content of tea polyphenols, theanine, caffeine, free amino acid, flavone, and soluble sugar, with 3 replicates per treatment. The specific testing methods were briefly described as follows: the tea leaves collected were fixed at 105 °C for 15 min, dried at 80 °C to constant weight, ground, and passed through 60 mesh sieves for the determination of the quality indicator of tea leaves. The determination of tea polyphenol content was referred to the national standard of the People’s Republic of China, “Determination of total polyphenols and catechins content in tea (GBT 8313-2018)”, and was determined via the folinol colorimetry method [42]. Theanine content was determined via high-performance liquid chromatography (HPLC) with reference to the national standard of the People’s Republic of China, “Determination of theanine in tea-Using high-performance liquid chromatography (GBT 23193-2017)” [43]. The caffeine content was determined with reference to the national standard of the People’s Republic of China, “Tea-Determination of caffeine content (GBT 8312-2013)”, and was determined via ultraviolet spectrophotometry [44]. The amino acid content was determined with reference to the national standard of the People’s Republic of China, “Tea-Determination of free amino acids content (GBT 8314-2013)”, and was determined via ninhydrin spectrophotometric method [45]. Flavonoid and soluble sugar contents were determined via aluminum trichloride colorimetric method and anthrone colorimetric method with reference to the method of Wang et al. [6], respectively.

### 3.4. Transcriptome Analysis of Tea Leaves

#### 3.4.1. RNA Isolation and Qualification

RNA was extracted using the TRIzol method (TIANGEN BIOTECH, Beijing, China) and treated with RNase-free DNase I (TaKaRa, Beijing, China). RNA degradation and contamination were monitored on 1% agarose gels. RNA was quantified using Agilent 2100 Bioanalyzer (Agilent Technologies, Santa Clara, CA, USA) and assessed for quality and integrity by NanoDrop spectrophotometer (IMPLEN, Westlake Village, CA, USA).

#### 3.4.2. Library Preparation for Transcriptome Sequencing

A total amount of 1.5 μg RNA per sample was used as input material for the RNA sample preparations. Sequencing libraries were generated using NEBNext^®^ Ultra™ RNA Library Prep Kit for Illumina^®^ (NEB, Ipswich, MA, USA) following the manufacturer’s recommendations, and indicator codes were added to assign sequences to each sample. Briefly, mRNA was purified from total RNA using poly-T oligo-attached magnetic beads. Fragmentation was performed using divalent cations at elevated temperatures in NEBNext First Strand Synthesis Reaction Buffer (5×). The first strand of cDNA was synthesized using a random hexamer primer and M-MuLV Reverse Transcriptase (RNase H). Second-strand cDNA synthesis was then performed using DNA Polymerase I and RNase H. Remaining overhangs were converted into blunt ends through exonuclease/polymerase activities. After adenylation of 3′ ends of DNA fragments, the NEBNext Adaptor with hairpin loop structure was ligated to prepare for hybridization. In order to select cDNA fragments preferentially 200–250 bp in length, library fragments were purified with the AMPure XP system (Beckman Coulter, Beverly, CA, USA). Then, 3 μL USER Enzyme (NEB, USA) was used with size-selected, adaptor-ligated cDNA at 37 °C for 15 min followed by 5 min at 95 °C before PCR. Then, PCR was performed with Phusion High-Fidelity DNA polymerase, Universal PCR primers, and Indicator (X) primers. Finally, PCR products were purified (AMPure XP system), and library quality was assessed on an Agilent Bioanalyzer 2100 system. Library preparations were sequenced on an Illumina Hiseq 4000 platform by Beijing Allwegene Technology Company Limited (Beijing, China), and paired-end 150 bp reads were generated.

### 3.5. Data Analysis

Quality control: Raw data (raw reads) with fastq format were first processed via internal perl scripts. First, Trimmomatic software (v0.33) was used to remove reads containing adapters, and then reads with N-containing ratios greater than 10% were removed, and reads in which the number of bases with a quality value of Q < 20 accounted for more than 50% of the entire read were removed to obtain lean data (clean reads). At the same time, Q20, Q30, GC content, and sequence repeat levels were calculated for clean data. All downstream analysis was based on clean data with high quality [46].

Mapping analysis: Adaptor sequences and low-quality sequence reads were removed from the dataset. Raw sequences were transformed into clean reads after data processing. These clean reads were then mapped to the reference genome sequence via STAR. Only reads with a perfect match or mismatch were further analyzed and annotated based on the reference genome.

Quantification of gene expression levels: HTSeq v 0.5.4 p3 was used to calculate the number of reads mapped to each gene. Gene expression levels were estimated by fragments per kilobase of transcript per million mapped fragments (FPKM) [47].

Difference analysis: DESeq2 v1.22.1 was used to analyze the differential expression between the two treatments, and the P value was corrected using the Benjamini & Hochberg method. The corrected P value and |log2foldchange| were used as the threshold for significant difference expression [48].

Key gene screening: The orthogonal partial least squares discrimination analysis (OPLS-DA) model between two treatments was constructed based on the acquisition of differential genes [15]. The variable importance for projection (VIP) of the OPLS-DA model was calculated to measure the influence of expression patterns of each gene on the classification and discrimination of the two samples and the explanatory ability to obtain the key genes of VIP > 1.

### 3.6. Statistical Analysis

Excel 2017 software was used to calculate the mean value and variance of the data. Rstudio 3.3 software was used to make cloud and rain plots, principal component plots, volcanic plots, OPLS-DA models (Package used in this regard was ropls and mixOmics), box plots, and trend plots. Based on the Pearson correlation coefficients between different indicators, Cytoscape_v3.9.1 software was used to produce a correlation network plot.

## 4. Conclusions

In this study, we analyzed the effect of pruning on growth indicators, quality indicators, and leaf gene expression of tea plants. The results showed (Figure 6) that pruning increased the gene expression of fatty acid synthesis, carbohydrate metabolism, nitrogen metabolism, protein processing in endoplasmic reticulum and plant hormone signaling pathways in tea plant leaves, and increased the chlorophyll content, leaf area, hundred bud weight and yield of tea plant, but reduced the gene expression of biosynthesis of secondary metabolites, flavonoid biosynthesis, phenylpropanoid biosynthesis and sesquiterpenes pathways in tea plant leaves, and reduced the content of caffeine, flavonoids and free amino acid of tea plant leaves. Therefore, it was assumed that after tea pruning, tea plants responded to abiotic stress not by increasing the synthesis and accumulation of secondary metabolites to resist the external environment but by increasing their own growth and thus improving their own resistance to the effect of pruning on tea plants. In conclusion, this study found that pruning was beneficial to promoting the growth of tea plants and increasing tea yield but was not conducive to the accumulation of some quality indicators in tea, especially caffeine, flavonoids, and free amino acids, which, in turn, reduced the quality of tea. After pruning, the tea plant was mainly growth-oriented. Therefore, in the process of tea plantation management, the tea plant pruned should be appropriate to increase the amount of fertilizer in order to meet the needs of tea plant growth. At the same time, we should pay attention to tea quality to avoid improving tea yield but reducing tea quality. This study provides an important theoretical reference for the management of agronomic measures in tea plantations. However, this study focused on the key genes that were mainly changed in tea leaves after pruning, and of course, there were some genes, the expression of which was also changed. The various genes could be further examined to investigate the mechanisms behind alterations in tea yield and quality resulting from pruning.

## Figures and Tables

**Figure 1 plants-12-03625-f001:**
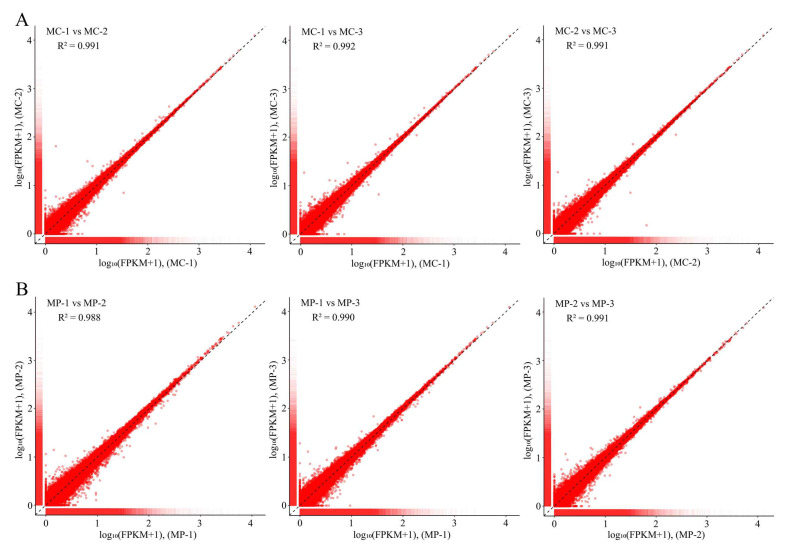
Repeatability test of transcriptome sequencing results of the tea plant. Note: MC: Tea plant unpruned; MP: Tea plant pruned; (**A**): Similarity analysis of gene expression in three biological repeats of MC; (**B**): Similarity analysis of gene expression in three biological repeats of MP.

**Figure 2 plants-12-03625-f002:**
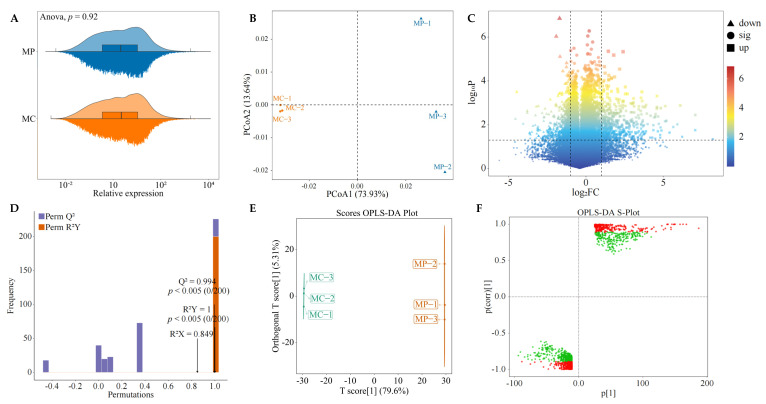
Gene expression analysis of tea plant under different treatments. Note: MC: Tea plant unpruned; MP: Tea plant pruned; (**A**): Analysis of total gene expression of tea plant under different treatments; (**B**): PCoA analysis of different treatments based on gene expression level; (**C**): Volcanic map screening of differentially expressed genes between different treatments; (**D**): OPLS-DA model test of different treatments; (**E**): Score chart of OPLS-DA model of different treatments; (**F**): Screening of key genes differentially expressed between treatments.

**Figure 3 plants-12-03625-f003:**
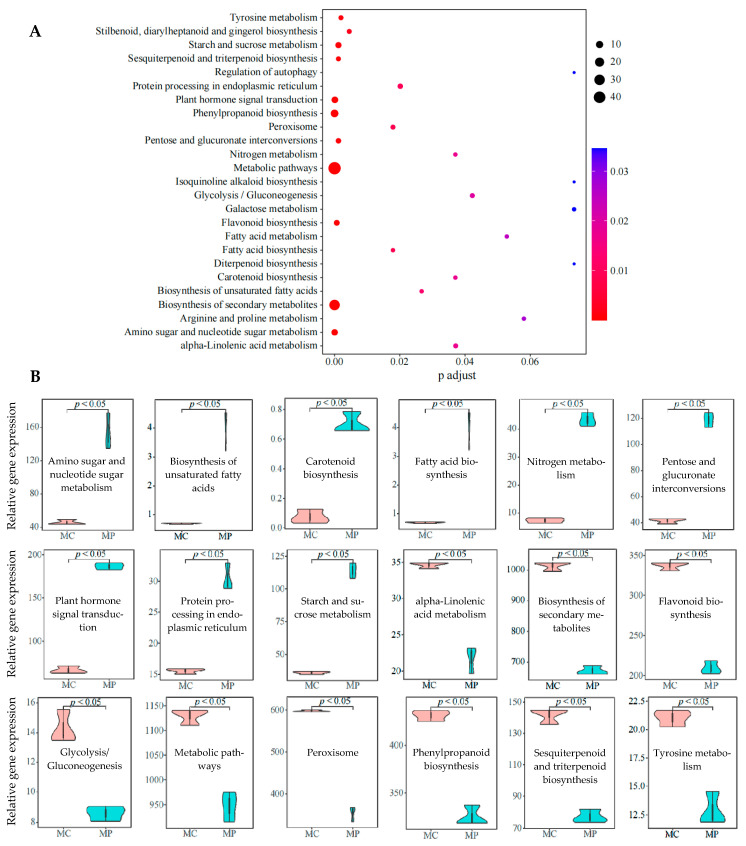
KEGG pathway enrichment and gene expression analysis of key genes. Note: MC: Tea plant unpruned; MP: Tea plant pruned; (**A**): KEGG pathway enrichment of key genes; (**B**): Gene expression analysis of metabolic pathways significantly enriched in KEGG.

**Figure 4 plants-12-03625-f004:**
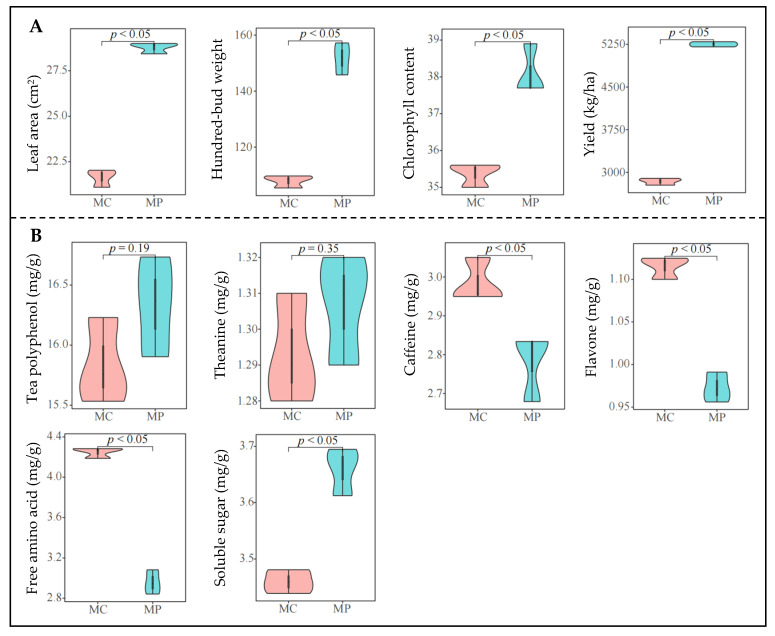
Analysis of the growth and quality indicator of tea plant. Note: MC: Tea plant unpruned; MP: Tea plant pruned; (**A**) analysis of the growth indicator of tea plant; (**B**) analysis of the quality indicator of tea plant.

**Figure 5 plants-12-03625-f005:**
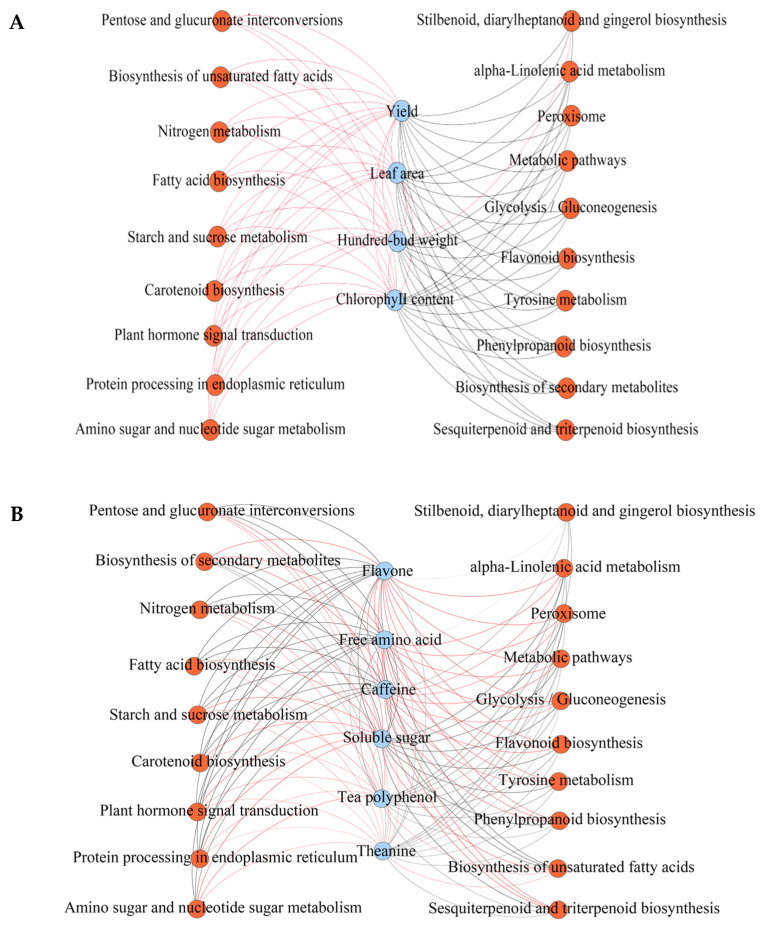
Analysis of the interaction between growth indicators, quality indicators, and gene expression of metabolic pathway in tea plant. Note: MC: Tea plant unpruned; MP: Tea plant pruned; The red line indicates a positive correlation, and the black line indicates a negative correlation; (**A**) interaction between growth indicators and gene expression levels of metabolic pathway in tea plant; (**B**) interaction between quality indicators and gene expression levels of metabolic pathway in tea leaves; The interaction network was analyzed as a correlation between the intensity of gene expression of different metabolic pathways and the growth and quality indexes of the tea tree, and Pearson correlation coefficients of positive or negative indicated positive or negative correlation, respectively.

**Figure 6 plants-12-03625-f006:**
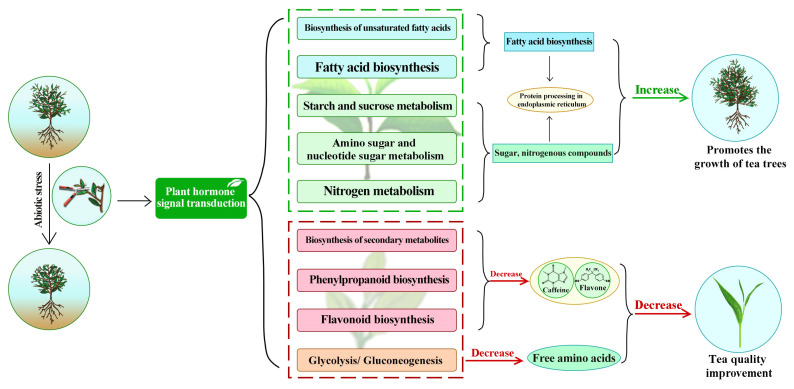
Transcriptome mechanism analysis of effects of pruning on growth and quality of tea plants.

## Data Availability

The original contributions presented in the study were publicly available. The transcriptome data can be found here: NCBI, PRJNA943532 (https://submit.ncbi.nlm.nih.gov/subs/sra/SUB12941038/overview, accessed on 8 September 2023).

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
