# Peer review of "Transcriptomic Analysis of the Effect of Pruning on Growth, Quality, and Yield of Wuyi Rock Tea"

_plants, 2023, doi:10.3390/plants12203625_

Round 1
Reviewer 1 Report
This study investigated the effects of pruning on gene expression, as well as the impact on the growth and quality of tea plants, through transcriptome analysis. The results showed significant changes in gene expression after pruning, and key genes and metabolic pathways were identified. Additionally, measurements were taken on the growth and quality indicators of the tea plants, and the correlation between these indicators and gene expression was analyzed. The findings indicated that pruning can promote tea plant growth and increase tea yield, but it may have negative effects on certain quality indicators, such as the accumulation of caffeine and flavonoids. By combining transcriptome analysis with measurements of growth and quality indicators, this study provides a comprehensive explanation for a deeper understanding of the effects of pruning on tea plants. The structure of the paper is logical, and the writing is smooth.
Major comments:
1、The description of the analysis methods for growth and quality indicators is unclear. More details and explanations of the application of statistical tests should be provided.
2、The article would benefit from a more elaborate discussion on the implications of the results. How do the research findings contribute to the existing knowledge in the field? What practical significance do they hold for tea plantation management?
3、A more comprehensive explanation of the identified key genes and metabolic pathways should be provided. What are the functions of these genes and pathways? How do they relate to the observed changes in growth and quality indicators?
4、This study only considers one location and one variety, overlooking the influence of factors such as variety differences and geographical variations on tea quality. To obtain more comprehensive data, it would be beneficial to compare and analyze various aspects such as different tea plant varieties, growth environments and cultivation fields, seasons, and plant ages to explore the factors that affect tea plant metabolisms and the formation of tea quality.
Minor comments:
99 The X-axis title in Figure 1B for the correlation analysis of MP-2 and MP-3 expression levels is incomplete.
343 Currently, multiple software options are available for transcriptome data quality control. Why were self-written scripts used instead? What specific criteria were employed for quality control?
356 It is mentioned here that two software programs were used to identify differentially expressed genes. However, it is worth noting that different software options could yield significant differences in the results. What specific software was utilized in this study?
370 How are the key genes associated with the observed changes in growth and quality indicators? What specific test method was employed?
427 Although the paper includes a data availability statement, it appears that the ID format of the dataset is incorrect.
In general, this paper offers valuable insights into the influence of pruning on tea tree growth and quality. The study exhibits high scientific credibility and employs an appropriate experimental design. However, there is a need for improvement, such as elucidating the research question and discussing the statistical analysis methods and significance of the results.
Author Response
Comments and Suggestions for Authors
This study investigated the effects of pruning on gene expression, as well as the impact on the growth and quality of tea plants, through transcriptome analysis. The results showed significant changes in gene expression after pruning, and key genes and metabolic pathways were identified. Additionally, measurements were taken on the growth and quality indicators of the tea plants, and the correlation between these indicators and gene expression was analyzed. The findings indicated that pruning can promote tea plant growth and increase tea yield, but it may have negative effects on certain quality indicators, such as the accumulation of caffeine and flavonoids. By combining transcriptome analysis with measurements of growth and quality indicators, this study provides a comprehensive explanation for a deeper understanding of the effects of pruning on tea plants. The structure of the paper is logical, and the writing is smooth.
Major comments:
1、The description of the analysis methods for growth and quality indicators is unclear. More details and explanations of the application of statistical tests should be provided.
A: Thanks to the reviewers . The authors have made additional changes to make it more detailed. Hopefully, it will meet the requirements.
2、The article would benefit from a more elaborate discussion on the implications of the results. How do the research findings contribute to the existing knowledge in the field? What practical significance do they hold for tea plantation management?
A: Thanks to the reviewers . The authors have revised the text appropriately and also added the implications for the management of tea plantations in the conclusion.
3、A more comprehensive explanation of the identified key genes and metabolic pathways should be provided. What are the functions of these genes and pathways? How do they relate to the observed changes in growth and quality indicators?
A: Thanks to the reviewers. The authors have made appropriate revisions to detail the function of the pathways enriched by the genes and to illustrate their relationship with growth indexes and quality indexes through the interaction network.
4、This study only considers one location and one variety, overlooking the influence of factors such as variety differences and geographical variations on tea quality. To obtain more comprehensive data, it would be beneficial to compare and analyze various aspects such as different tea plant varieties, growth environments and cultivation fields, seasons, and plant ages to explore the factors that affect tea plant metabolisms and the formation of tea quality.
A: Thanks to the reviewers. The suggestions of the reviewing experts are very good. This manuscript focuses on the changes in the transcriptome of tea tree leaves and its relationship with growth and quality after different treatments of the same tea tree variety at the same site. Of course, the author's group is also studying two other varieties "Shuixian" and "Rougui", mainly from the perspective of soil root ecology, and also found that pruning is beneficial to promote the growth of the tea tree, but not conducive to the formation of quality. Related research has been published in "Microbiology Spectrum" and "Agronomy". The environment of tea tree cultivation and pruning season may affect the growth and quality of tea tree, which is under continuous research and exploration by our group. Thank you very much for your suggestion.
Minor comments:
99 The X-axis title in Figure 1B for the correlation analysis of MP-2 and MP-3 expression levels is incomplete.
A: Thank you to the reviewers. The authors have revised it.
343 Currently, multiple software options are available for transcriptome data quality control. Why were self-written scripts used instead? What specific criteria were employed for quality control?
A: Thank you to the reviewers. The authors used Trimmomatic software (v0.33) for quality control, which has been added to the manuscript with additional parameter settings for quality control.
356 It is mentioned here that two software programs were used to identify differentially expressed genes. However, it is worth noting that different software options could yield significant differences in the results. What specific software was utilized in this study?
A: Thanks to the reviewers for their careful review. The software used by the authors to identify the differentially expressed genes was DESeq2 v1.22.1. This was an error on the part of the authors, and the authors have revised it. Thank you very much.
370 How are the key genes associated with the observed changes in growth and quality indicators? What specific test method was employed?
A: Thanks to the reviewing experts. Based on the pearson correlation coefficient between different indexes, the authors make the correlation network diagram. The authors have added it in the paper.
427 Although the paper includes a data availability statement, it appears that the ID format of the dataset is incorrect.
A: Thanks to the reviewing experts. The authors have revised it as follows: (PRJNA943532 (https://submit.ncbi.nlm.nih.gov/subs/sra/SUB12941038/overview).
In general, this paper offers valuable insights into the influence of pruning on tea tree growth and quality. The study exhibits high scientific credibility and employs an appropriate experimental design. However, there is a need for improvement, such as elucidating the research question and discussing the statistical analysis methods and significance of the results.
A: Thanks to the reviewing experts. The authors have carefully revised the manuscript as requested by the experts. Thank you very much.
Reviewer 2 Report
The manuscript tilted" Transcriptomic analysis of the effect of pruning on growth and quality of Wuyi rock tea plants".
The idea of the manuscript is very good and it is very important for the tea growers. The paper can show these results for the large farmers and for the tea factories. The authors used the transcriptome method to prove their idea and I see that is not completely enough and will not be completely satisfy for the growers or the industry. So, I have some afew comments on the manuscript.
The title should be modified into" Transcriptomic analysis of the effect of pruning on growth and both of quality and quantity of Wuyi rock tea plants ingredients.
Abstract: is too long and should be summarized more than its present state.
Introduction: is short and it contains a low number of citation and this low number of references will effect on the quality of article, especially the authors did not included anything about the transcriptome analysis and it's important to discover the phyiscological status of the plant under different kind of stresses. Also, authors should include other studies work on the same plant or another plant to ensure their idea and to reach their target.
Results and Discussion: contain a low number of citations which affect the interpretation of the obtained results. This deficiency of the interpretation affects the manuscript quality.
Figure 4 and 5 could be combined in one figure; this will represent the results more good or delete the figure 4, because figure 5 is enough.
Line 301, Determination of tea quality indicator: The authors made the analysis for the polyphenols, falvonoides but they forget the analysis of the antioxidant. Also, authors, should use in their comparative plant leaf samples from plants previously pruning since enough period to see if the plant able to recovery from the purning stress after a distinct time or not? This will ensure that the effect of pruning is dominant or instant.
Materials and methods:
Is very good and it well planned.
Conclusion:
Is too long and should be summarized.
Author Response
Comments and Suggestions for Authors
The manuscript tilted" Transcriptomic analysis of the effect of pruning on growth and quality of Wuyi rock tea plants".
The idea of the manuscript is very good and it is very important for the tea growers. The paper can show these results for the large farmers and for the tea factories. The authors used the transcriptome method to prove their idea and I see that is not completely enough and will not be completely satisfy for the growers or the industry. So, I have some afew comments on the manuscript.
The title should be modified into" Transcriptomic analysis of the effect of pruning on growth and both of quality and quantity of Wuyi rock tea plants ingredients.
A: Thank you to the reviewers. The authors have revised it.
Abstract: is too long and should be summarized more than its present state.
A: Thank you to the reviewers. The authors have revised the abstract to summarize the results obtained from the study.
Introduction: is short and it contains a low number of citation and this low number of references will effect on the quality of article, especially the authors did not included anything about the transcriptome analysis and it's important to discover the phyiscological status of the plant under different kind of stresses. Also, authors should include other studies work on the same plant or another plant to ensure their idea and to reach their target.
A: Thank you to the reviewers. The authors have added relevant content and cited references in the introduction.
Results and Discussion: contain a low number of citations which affect the interpretation of the obtained results. This deficiency of the interpretation affects the manuscript quality.
A: Thank you to the reviewers. The authors have made appropriate revisions and also added some references.
Figure 4 and 5 could be combined in one figure; this will represent the results more good or delete the figure 4, because figure 5 is enough.
A: Thank you to the reviewers. Figure 4 shows the analysis of the results of growth indexes. Figure 5 shows the correlation analysis between different indexes. The analysis is not the same. The authors recommend retaining Figures 4 and 5.
Line 301, Determination of tea quality indicator: The authors made the analysis for the polyphenols, falvonoides but they forget the analysis of the antioxidant. Also, authors, should use in their comparative plant leaf samples from plants previously pruning since enough period to see if the plant able to recovery from the purning stress after a distinct time or not? This will ensure that the effect of pruning is dominant or instant.
A: Thank you to the reviewers. The authors focused on the effect of pruning on the yield and quality of tea, so antioxidant was not determined in the manuscript. Of course, the change and recovery process of tea tree resistance after pruning is a very important process. The authors will analyze the dynamics of tea tree resistance after pruning in depth in subsequent studies. Thank you very much for your suggestions.
Materials and methods:
Is very good and it well planned.
A: Thank you to the reviewers.
Conclusion:
Is too long and should be summarized.
A: Thank you to the reviewers. The authors have revised the conclusions appropriately.
Reviewer 3 Report
I checked your manuscript and described comments below.
This paper provides a very good transcriptome analysis of the effects of pruning on Wuyi rock tea plants native to China.
Most people in the world don't know Wuyi rock tea. Wuyi rock tea is a kind of Oolong tea and is said to be the prototype of black tea. I think that this content will have a greater impact if it is described in this paper.
The following papers should also be included in the references of this paper.
Front. Plant Sci., Sec. Plant Metabolism and Chemodiversity, Volume 14 – 2023, doi: 10.3389/fpls.2023.1235687.
I don't think there is a specific problem with the analysis method.
I don't think this paper has any major mistakes or grammatical problems.
Author Response
Comments and Suggestions for Authors
I checked your manuscript and described comments below.
This paper provides a very good transcriptome analysis of the effects of pruning on Wuyi rock tea plants native to China.
Most people in the world don't know Wuyi rock tea. Wuyi rock tea is a kind of Oolong tea and is said to be the prototype of black tea. I think that this content will have a greater impact if it is described in this paper.
A: Thank you to the reviewer. The production process of Wuyi Rock Tea is semi-fermentation, and according to the classification of fermentation degree, Wuyi Rock Tea belongs to Oolong Tea. The author has made appropriate descriptions in the text.
The following papers should also be included in the references of this paper.
Front. Plant Sci., Sec. Plant Metabolism and Chemodiversity, Volume 14 – 2023, doi: 10.3389/fpls.2023.1235687.
A: Thank you to the reviewer. The authors have revised and made additions.
I don't think there is a specific problem with the analysis method.
A: Thank you to the reviewer.
I don't think this paper has any major mistakes or grammatical problems.
A: Thank you to the reviewer.
Reviewer 4 Report
Dear Editor,
I am writing in regard to the manuscript "Transcriptomic analysis of the effect of pruning on growth and quality of Wuyi rock tea plants" which I submitted for consideration for publication in Plants Journal.
The recommendation that this work should be repeated for 2 consecutive years to strengthen the conclusions is an important one that I have taken into consideration.
However, I remain convinced that the transcriptomic insights reported here will be of interest to researchers in this field after expansion to a multi-year study. I plan to carry out additional experiments over the next 2-3 growing seasons to generate more comprehensive data on the effects of pruning regimes on tea quality and growth morphology.
Best regards,
Author Response
Comments and Suggestions for Authors
I am writing in regard to the manuscript "Transcriptomic analysis of the effect of pruning on growth and quality of Wuyi rock tea plants" which I submitted for consideration for publication in Plants Journal.
The recommendation that this work should be repeated for 2 consecutive years to strengthen the conclusions is an important one that I have taken into consideration.
A: Thank you to the reviewer. The authors are continuing their research in this area. The current research in the manuscript is our preliminary results obtained. Your suggestions are much appreciated.
However, I remain convinced that the transcriptomic insights reported here will be of interest to researchers in this field after expansion to a multi-year study. I plan to carry out additional experiments over the next 2-3 growing seasons to generate more comprehensive data on the effects of pruning regimes on tea quality and growth morphology.
A: Thanks to the reviewer. Pruning is very important for the cultivation and management of Wuyi Rock Tea tea trees. Therefore the author's team has been continuously working on this area. The author's team has already conducted in-depth discussions on morphology, physiology, transcription, metabolism, proteome, and soil metagenome. Some of his research articles have been published in "Journal of Hazardous Materials", "Microbiology Spectrum", "Frontiers in Plant Science" and "Agronomy". It is very exciting to know that the expert is also working on this area and I look forward to reading your excellent work.
Best regards,
Reviewer 5 Report
-The investigation have contribution for developing methodology of identification of tea plants responses to biotic and abiotic stress and improving their growth ability and resistance to the external environment after pruning and specific expression of genes and their role in metabolic pathways in tea leaves
-This study addressed to developing methods of identification effects of pruning and unpruning on gene expression, growth indicators, and quality indicators of tea leaves, and affect gene expression of metabolic pathways in tea leaves.
- The subject of the article is within scope of the journal.
- The title clearly and sufficiently reflect its content
- The aim of study should be pointed out as a second sentence (after introductory sentence) in abstract. You can use the second sentence already written by highlighting what the aim of the research was.
- Keywords are appropriate.
- Introduction related to the subject of investigation. On the end of introduction, should be clearly pointed out aim of investigation.
- The used experimental methods are adequate.
- Results are clearly presented clear and Figures are clear.
- The references are adequate and necessary.
- Conclusions based on presented results an clearly presented
REMARKS: From line 79 to 82 those two sentence more belongs to chapter Conclusion. if you do not want to include the conclusion in the chapter, these two sentences should be deleted from this paragraph.
The aim of the research should be written in the last paragraph of the introduction
In line 374 and 375 ned delete (Fig 6). The citting tables, fig. And authors is not allowed in the chapter conclusion.
Author Response
Comments and Suggestions for Authors
-The investigation have contribution for developing methodology of identification of tea plants responses to biotic and abiotic stress and improving their growth ability and resistance to the external environment after pruning and specific expression of genes and their role in metabolic pathways in tea leaves
-This study addressed to developing methods of identification effects of pruning and unpruning on gene expression, growth indicators, and quality indicators of tea leaves, and affect gene expression of metabolic pathways in tea leaves.
- The subject of the article is within scope of the journal.
- The title clearly and sufficiently reflect its content
- The aim of study should be pointed out as a second sentence (after introductory sentence) in abstract. You can use the second sentence already written by highlighting what the aim of the research was.
A: Thank you to the reviewer. The authors have made changes and additions to the introduction to emphasize the aim of the study.
- Keywords are appropriate.
- Introduction related to the subject of investigation. On the end of introduction, should be clearly pointed out aim of investigation.
A: Thank you to the reviewer. The authors have made changes and additions to the introduction to emphasize the aim of the study.
- The used experimental methods are adequate.
- Results are clearly presented clear and Figures are clear.
- The references are adequate and necessary.
- Conclusions based on presented results an clearly presented
A: Thank you to the reviewers.
REMARKS: From line 79 to 82 those two sentence more belongs to chapter Conclusion. if you do not want to include the conclusion in the chapter, these two sentences should be deleted from this paragraph.
The aim of the research should be written in the last paragraph of the introduction
A: Thank you to the reviewer. The authors have revised the paragraph.
In line 374 and 375 ned delete (Fig 6). The citting tables, fig. And authors is not allowed in the chapter conclusion.
A: Many thanks to the reviewers. The authors have placed Figure 6 here in the hope that it will give the reader a better understanding of the conclusions throughout the text. The authors suggest to keep it. Hopefully it will meet your requirements. Thank you very much for your suggestion.
Round 2
Reviewer 1 Report
My concerns have been resolved.
Reviewer 4 Report
.